# Track4Animate3D: Animating Any 3D Model via Multi-View Diffusion with Point-Tracking Motion Priors

## Abstract

Generating 4D objects is challenging because it requires jointly maintaining appearance and motion consistency across space and time under sparse inputs, while avoiding artifacts and temporal drift. We hypothesize that this view discrepancy stems from supervision that relies solely on pixel- or latent-space video-diffusion losses and lacks explicitly temporally aware tracking guidance at feature-level. To address this issue, we introduce *Track4Animate3D*, a two-stage framework that unifies a multi-view video diffusion model with a foundation point tracker and a hybrid 4D Gaussian Splatting (4D-GS) reconstructor. The core idea is to explicitly inject motion priors from a foundation point tracker into the feature representation for both video generation and 4D-GS. In Stage One, we impose dense, feature-level point correspondences within the diffusion generator, enforcing temporally consistent feature representations that suppress appearance drift and strengthen cross-view coherence. In Stage Two, we reconstruct a dynamic 4D-GS using a hybrid motion representation that concatenates co-located diffusion features (carrying tracking priors from Stage One) with Hex-plane features, and appends a 4D Spherical Harmonics modeling, improving higher-fidelity dynamics and illumination modeling. *Track4Animate3D* outperforms strong baselines (e.g., Animate3D, DG4D) across VBench metrics for multi-view video generation, CLIP-O/F/C metrics and user preference studies for 4D generation, producing temporally stable and text-editable 4D assets. Finally, we curate a new high-quality 4D dataset named *Sketchfab28*, to evaluate object-centric 4D generation for future research.

## 1 Introduction

4D assets generation aims to synthesize temporally coherent, animatable 3D assets that evolve over time, jointly modeling geometry, appearance, and motion from text, single images, monocular video, or static 3D model. Its applications span game/movie asset creation, AR/VR avatars, digital humans, and interactable simulation and data generation for robotics and autonomous driving. Most recent methods leverage deformable neural fields Mildenhall et al. (2021), dynamic 3D Gaussians Kerbl et al. (2023) and 4D mesh Rossignac (2002) as 4D representation, which first produce temporally consistent multi-view video from diffusion models, and then reconstruct/refine 4D assets including Score Distillation Sampling(SDS)-free and class-level priors. The difficulty of 4D generation stems from jointly maintaining appearance and dynamic motion spatiotemporal consistency across space and time under sparse inputs, while avoiding artifacts such as Janus effects and temporal drift.

Recent work on 4D object generation can be organized by conditioning modality, text, single image, monocular video, or static 3D model. Text to 4D methods, e.g., MAV3D Singer et al. (2023), AYG Ling et al. (2024) and CT4D Chen et al. (2024) leverage composed diffusion priors with dynamic 3D Gaussians to synthesize animatable objects with improved temporal coherence. Image to 4D approaches, e.g., DG4D Ren et al. (2023), EG4D Sun et al. (2025) and Animate124 Zhao et al. (2023), move beyond SDS by first producing temporally consistent multi-view video and then reconstructing/refining dynamic Gaussian assets. Monocular video to 4D pipelines, e.g., SV4D Xie et al. (2024), SC4D Wu et al. (2024b), DreamMesh4D Li et al. (2024) and Consistent4D Jiang et al. (2024b), decouple motion from appearance or bind Gaussians to mesh surfaces to achieve editable, temporally consistent dynamic objects. 3D to 4D techniques, e.g., Animate3D Jiang et al. (2024a),

HyperDiffusion Erkoç et al. (2023) and ElastoGen Feng et al. (2024), animate static meshes by conditioning multi-view video diffusion on object renderings and coupling it with 4D-SDS for motion fidelity. Collectively, these directions highlight a shift toward object-centric priors and animation-friendly representations such as dynamic Gaussians and 4D mesh. However, the key challenges still remain in maintaining view–time coherence under sparse inputs, preserving identity and accurate geometry, achieving fine-grained controllability, and reducing optimization cost.

Most closely related to our *Track4Animate3D* is Animate3D Jiang et al. (2024a), which animates arbitrary static 3D models by conditioning a multi-view video diffusion model (MV-VDM) on multi-view renderings to synthesize temporally consistent videos, followed by 4D reconstruction and 4D-SDS refinement based on unified MV-VDM. Although foundation MV-VDM generators produce high-quality images, appearance drift still remains a persistent issue, with objects degrading or changing inconsistently across frames over time. We attribute this limitation stems from supervision that relies solely on video-diffusion losses in pixel/latent space, lacking explicit temporal aware tracking supervision at feature level. Among recent foundation trackers, CoTracker3 Karaev et al. (2024) achieves robust frame-to-frame multi-point tracking via cross-track attention, with strong occlusion handling. To overcome the limitations of Animate3D, we introduce *Track4Animate3D* to inject dense frame-to-frame point correspondences into the multi-view video diffusion model with stronger temporal supervision on diffusion features, by leveraging the foundational tracking capability of CoTracker3 Karaev et al. (2024).

Concretely, *Track4Animate3D* follows a two-stage architecture comprising (i) a foundation 4D multi-view video generator and (ii) a joint 4D Gaussian Splatting (4D-GS) reconstruction. In Stage One, we upgrade the 4D multi-view video generator with foundation keypoint tracking. We first use CoTracker3 Karaev et al. (2024) to obtain dense keypoint tracks on ground-truth videos and then locate their counterparts in the diffusion feature space of the generator. A Correspondence Tracking Loss is utilized to align these feature-level matches across frames, providing explicit temporal supervision that suppresses appearance drift and improves cross-frame coherence. In Stage Two, we jointly optimize 4D-GS Wu et al. (2024a) with photometric/reconstruction losses and 4D-SDS driven by the multi-view videos from Stage One. Unlike classic 4D-GS Wu et al. (2024a) pipelines that learn motion with only Hex-plane Fridovich-Keil et al. (2023) features, we augment each 3D sample point along with time by concatenating Hex-plane features with its co-located Stage One diffusion features. These diffusion features, trained under CoTracker supervision, encode motion priors that strengthen point representations of dynamic geometry and appearance. Additionally, we introduce a 4D Spherical Harmonics (SH) modelling into 4D-GS, which improves illumination modeling of 4D assets.

We highlight our main contributions as follows:

- We introduce *Track4Animate3D*, a novel 4D generation framework that unifies diffusion generator, foundation point tracker, and 4D-GS, directly leveraging motion priors from foundation tracking to mitigate appearance drift via explicit temporal supervision.

- We propose a temporal-aware multi-view 4D generator combining video-diffusion objectives with cross-frame dense point tracking/correspondence, enhancing the temporal consistency of diffusion feature representations.

- We develop a joint 4D-GS reconstruction that learns motion field from hybrid representations by concatenating co-located diffusion features (carrying tracking priors) with Hex-plane features for each spatio-temporal sample, yielding higher-fidelity dynamics and appearance. We also introduce a 4D Spherical Harmonics modeling into 4D-GS to improve illumination modeling of 4D assets.

- We construct a new high-quality 4D dataset, *Sketchfab28*, to support object-centric 4D generation evaluation for further research.

## 2 RELATED WORK

**Text-to-4D** Early pipelines adapted SDS from text-to-3D/video to animate objects over time. AYG Ling et al. (2024) composes diffusion priors with dynamic 3D Gaussians, enabling efficient text-conditioned 4D assets. CT4D Chen et al. (2024) generates animatable meshes through a staged text→3D→motion pipeline for better controllability. Recent directions strengthen supervision or

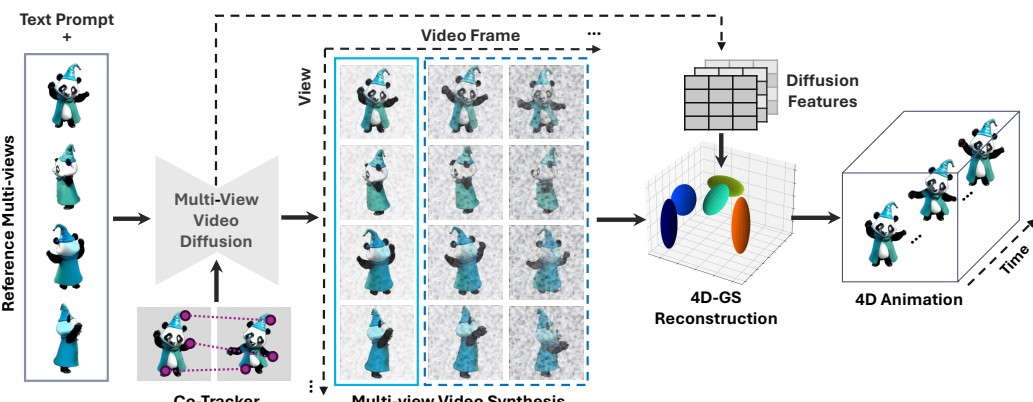

Figure 1: The *Track4Animate3D* pipeline comprises two stages: 1) multi-view video generation and 2) 4D-GS reconstruction.

structure to mitigate drift and improve temporal fidelity, e.g., PLA4D Miao et al. (2024) (pixel-aligned supervision) and Trans4D Zeng et al. (2024) (state-transition control).

**Single Image-to-4D** Given a single image, two strategies for 4D generation dominate: (i) SDS-based image/video diffusion guidance (e.g., Animate124 Zhao et al. (2023)); and (ii) explicit supervision via short generated multi-view videos for reconstruction, improving efficiency and consistency (e.g., DG4D Ren et al. (2023), EG4D Sun et al. (2025)). Diffusion[2] further unifies multi-view/video diffusion priors to boost view–time consistency Yang et al. (2025). These methods commonly reconstruct dynamic Gaussian assets from synthesized frames.

**Monocular Video-to-4D** From a single-view input video, recent works enhance view–time coherence via priors and structure: SV4D Xie et al. (2024) trains multi-frame/multi-view consistency with SDS; SC4D Wu et al. (2024b) uses sparse control points and skinning to decouple motion/appearance and enable motion transfer; DreamMesh4D Li et al. (2024) binds Gaussians to mesh faces for editable, high-fidelity assets; Consistent4D Jiang et al. (2024b) emphasizes 360° consistency with alignment losses and curated data; 4Diffusion Zhang et al. (2024) learns a multi-view video diffusion prior to supervise dynamic 4D reconstruction.

**Static 3D-to-4D** Methods animate a provided mesh by conditioning multi-view video diffusion on multi-view renderings to synthesize motion, then reconstruct/refine a deformable 4D representation (e.g., Animate3D Jiang et al. (2024a)). Alternatives generate motion by weight-space diffusion over implicit fields (HyperDiffusion Erkoç et al. (2023)) or by learned elastodynamics for physically plausible deformation (ElastoGen Feng et al. (2024)).

## 3 METHODOLOGY

### 3.1 OVERVIEW

Given an off-the-shelf static 3D model, our aim is to animate it from a text prompt while using its multi-view static renderings as image condition. As shown in Fig. 1, *Track4Animate3D* follows a two-stage pipeline: 1) we learn a multi-view video diffusion model that, conditioned on static multi-view images rendered from the 3D static object and the text description, synthesizes temporally coherent multi-view videos; 2) we learn 3D object animation via a 4D-GS representation, taking the static 3D Gaussians as the canonical asset and estimating time-dependent motion fields from the generated multi-view videos. We leverage the motion prior from a foundation tracking model to strengthen both stages. Stage One establishes temporally consistent diffusion features for video generation via dense, correspondence-aware point-tracking supervision. Stage Two converts the synthesized multi-view videos into a dynamic 4D Gaussian representation, enriched by tracking-aware diffusion features.

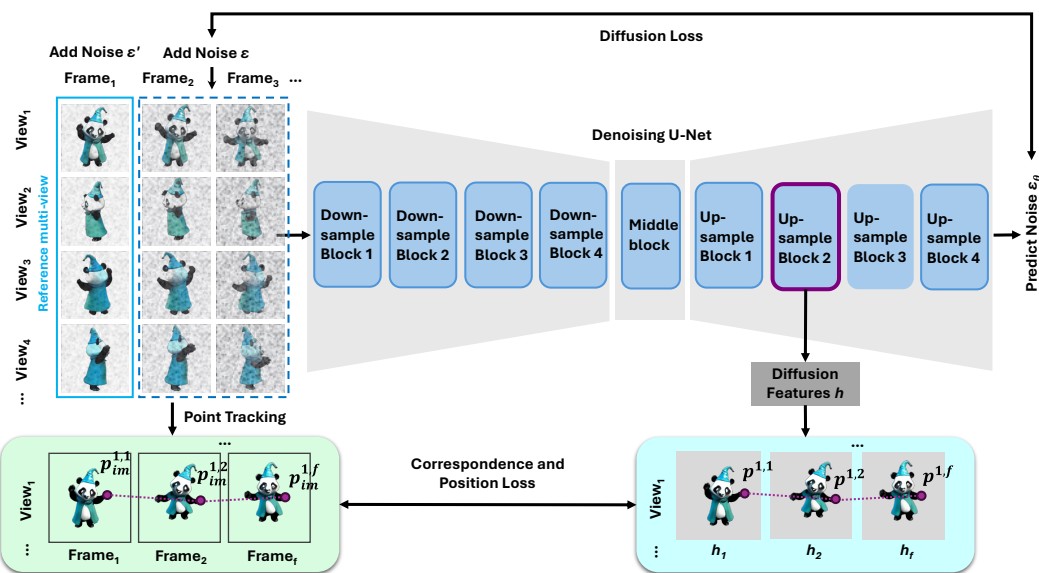

Figure 2: The multi-view video diffusion with dense point tracking.

### 3.2 MULTI-VIEW VIDEO DIFFUSION WITH DENSE POINT TRACKING

Beyond classic MV-VDM Jiang et al. (2024a), we couple multi-view diffusion with a foundation tracker, injecting dense feature-level motion priors to enforce correspondence-aware temporal consistency for multi-view video generation, as shown in Fig 2.

**Network Architecture:** We build on the MV-VDM backbone Jiang et al. (2024a), enrich cross-view diffusion with the multi-view 3D attention of MVDream Shi et al. (2023), and adopt the temporal attention of AnimateDiff Guo et al. (2023). The network follows a standard denoising U-Net with an encoder and decoder: the encoder comprises four spatiotemporal downsample blocks, the decoder mirrors them with four spatiotemporal upsample blocks, and a middle block bridges the downsample and upsample. Each spatiotemporal block comprises three main modules: i) Multi-view 3D Attention, which injects view-indexed tokens and calibrated camera embeddings into the U-Net to aggregate 3D geometry-consistent evidence across different views; ii) an MV2V Adapter that concatenates noisy frames along the spatial dimension to query the rich contextual information from the multi-view conditional frames; and iii) Spatiotemporal Attention that couples spatial attention to extract frame-wise features from different views, and temporal attention to model motion.

Our multi-view video diffusion training follows the Latent Diffusion Model Rombach et al. (2022), operating in the low-dimensional latent space of a pretrained U-Net. Given video image $x$ from $N$ camera views and $F$ frames, the U-Net denoises a latent tensor $z \in \mathbb{R}^{N \times F \times C \times H \times W}$ with timestep $t$ using view/time-aware self-attention and cross-attention over camera $c$ and text prompts $y$, producing multi-view video samples consistent across both space and time.

**Diffusion Features for Point Tracking:** Prior work Yang & Wang (2023); Yu et al. (2024); Xiang et al. (2023); Jeong et al. (2025) show that diffusion models encode discriminative, task-useful representations in their hidden states. Inspired by this, we seek the most stable feature space for point tracking in object-centric multi-view videos, i.e., features that reliably encode long-range temporal correspondences. We therefore probe the long-term tracking capability of U-Net–based video diffusion models by evaluating, block-by-block, which features best support correspondence. Concretely, for a real video we add slight noise, run the denoiser, extract layer-wise feature maps from every block, and compute tracks by cosine-similarity nearest-neighbor matching Tang et al. (2023); Luo et al. (2023) for a fixed set of query points on the first frame, as shown in Fig. 3. This analysis yields a consistent finding: features from the *second spatiotemporal upsampling block* in the decoder of U-Net provide the strongest temporal correspondences, in particular those from the *temporal motion module* within the Spatiotemporal Attention block.

Based on this finding, we uniformly sample query points from the first-frame multi-view images. Specifically, for each view we overlay a $15 \times 15$ grid and discard candidates outside the object using

Figure 3: Left: points tracking in multi-view images; Right: tacked points in similarity heatmap of U-Net's second Upsample Block. Brighter colors indicate higher feature similarity.

the instance mask. During training, we randomly choose 8 points per view ($8 \times n$ tracking points in total) from the query points set, yielding 8 tracked points in each view for the first frame, and use CoTracker3 Karaev et al. (2024) to track their trajectories across the multi-view video. Leveraging the correspondence/mapping between pixel locations in the original image and coordinates on the U-Net's feature map, we then localize each tracked point in the diffusion features of the second spatiotemporal upsampling block in U-Net's decoder. Finally, we extract the corresponding feature descriptors from this diffusion feature space for downstream supervision. *Track4Animate3D* leverages point tracking at feature level as auxiliary supervision to strengthen the temporal awareness of video-diffusion features.

**Training Objectives:** Given sampled multi-view video frames $x_0^{1:n,1:f}$ across views $[1:n]$, and frames $[1:f]$, we first encode them with the encoder $\mathcal{E}$ into latent feature $z_0^{1:n,1:f} = \mathcal{E}(x_0^{1:n,1:f})$ from $n$ camera views and $f$ frames. Gaussian noise $\epsilon$ is then injected to frames $2:f$ by the forward diffusion scheduler as,

$$z_t^{1:n,\,2:f} = \sqrt{\overline{\alpha}_t}\, z_0^{1:n,\,2:f} + \sqrt{1 - \overline{\alpha}_t}\, \epsilon, \ \ \epsilon \sim \mathcal{N}(0, \mathbf{I}) \tag{1}$$

where $\overline{\alpha}_t$ is a weighted noise scale. To encourage sufficient motion dynamics during 4D generation, we also inject a time dependent noise $\epsilon'$ into the first frame (the multi-view conditional frames), following Zhao et al. (2024),

$$z_t^{1:n,\,1} = z_0^{1:n,\,1} + \beta_t \epsilon', \ \ \epsilon' \sim \mathcal{N}(0, \mathbf{I}) \tag{2}$$

where $\beta_t$ denotes the noise scale. This scheduling principle is to apply larger noise at large timesteps to sufficiently perturb $z_0^{1:n,\,1}$, and progressively anneal the noise as the timestep decreases. During training, the denoiser takes as input the conditional latent $z_t^{1:n,\,1}$, noisy latent code $z_t^{1:n,2:f}$, text prompt embedding $y$, and the camera parameters $c^{1:n}$ across views, and predicts the noise. The diffusion loss $\mathcal{L}_{\text{diff}}$ is defined as,

$$\mathcal{L}_{\text{diff}} = \mathbb{E}_{\mathcal{E}(x_0),\, y,\, \epsilon \sim \mathcal{N}(0,\mathbf{I}),\, t} \left[ \left\| \epsilon - \epsilon_\theta\big(z_t^{1:n,1},\, z_t^{1:n,2:f},\, t,\, y,\, c^{1:n}\big) \right\|_2^2 \right], \tag{3}$$

where $\theta$ denotes the diffusion U-Net. Consistent with Animate3D Jiang et al. (2024a), we freeze the entire multi-view 3D attention module and train only the MV2V-Adapter and spatiotemporal attention blocks to reduce memory and speed up optimization.

*Track4Animate3D* leverages a foundation point tracker to impose auxiliary, feature-level point-tracking supervision on video-diffusion features, thereby strengthening temporal awareness. Using the mapping between pixel locations $\{p_{im}^{i,j}\}$ in images and coordinates $\{p^{i,j}\}$ on diffusion feature maps, we localize all the tracked points $p^{i,j}$ on the feature maps across views $i \in [1:n]$ and frames $j \in [1:f]$ by bilinear interpolation, and extract their hidden descriptors $h(p^{i,j})$. We then utilize a correspondence loss $\mathcal{L}_{\text{corr}}$ that enforces consistency of tracked points' hidden-space descriptors in two adjacent frames, implemented as a cosine-similarity objective,

$$\mathcal{L}_{\text{corr}} = \frac{1}{nf} \sum_{i=1}^{n} \sum_{j=1}^{f-1} \big(1 - \cos\_\text{sim}\big(h(p^{i,j}),\, h(p^{i,j+1})\big)\big). \tag{4}$$

We additionally estimate the tracked locations $\hat{p}^{1:n,\,1:f}$ directly in the diffusion feature space using a cosine-similarity nearest–neighbor search, implemented as a soft-argmax over the cosine-similarity

Figure 4: 4D-GS reconstruction with hybrid feature representations.

map: $\hat{p}^{i,j} = \text{soft\_argmax}(\text{cos\_sim}(\mathbf{h}^{i,j}, h(p^{i,1}))$. We impose a position loss $\mathcal{L}_{\text{pos}}$ to minimize the discrepancy between the predicted and tracked locations:

$$\mathcal{L}_{\text{pos}} = \frac{1}{nf} \sum_{i=1}^{n} \sum_{j=2}^{f} L_{\text{Huber}}\left( p^{i,j} - \hat{p}^{i,j} \right), \tag{5}$$

where $L_{Huber}$ denotes the Huber loss. The final objective of Stage One is $\mathcal{L}_1 = \lambda_1 \mathcal{L}_{\text{diff}} + \lambda_2 \mathcal{L}_{\text{corr}} + \lambda_3 \mathcal{L}_{\text{pos}}$ with $\lambda_1, \lambda_2, \lambda_3$ as weighting coefficients.

### 3.3 4D-GS RECONSTRUCTION WITH HYBRID FEATURE REPRESENTATIONS:

Compared to classic 4D-GS Wu et al. (2024a), we introduce two key innovations as shown in Fig. 4: 1) a hybrid motion representation fusing Stage One tracking-aware diffusion features and Hex-plane features; 2) a 4D Spherical Harmonics (SH) modeling.

**4D Motion Fields with Hybrid Representations:** We initialize the 3D-GS representation of the object by sampling Gaussians from the mesh's vertices and triangles. Each Gaussian is parameterized as static 3D Gaussian as $\mathcal{G}_{3D} = \{\mathcal{X}, \mathcal{C}_{3D}, \alpha, r, s\}$, where $\mathcal{X}$, $\mathcal{C}_{3D}$, $\alpha$, $r$, and $s$ represent the position, color, opacity, rotation, and scale, respectively. The motion module predicts per-frame changes in position, rotation, and scale for each Gaussian point in frame $f$ by interpolating the Hex-planes $\mathcal{H}$ and diffusion feature-planes $\mathcal{D}$ from the second upsampling block in U-Net's decoder. For a Gaussian center $\mathcal{X}$ at frame $f$, we form the hybrid feature by:

$$\mathcal{F} = \bigcup_{Hex} \prod_{\zeta_1} \text{interp}\left( \mathcal{H}^{\zeta_1}, (\mathcal{X}, f) \right) \oplus \bigcup_{Diff} \prod_{\zeta_2} \text{interp}\left( \mathcal{D}^{\zeta_2}, K[E](\mathcal{X}, f) \right) \tag{6}$$

where $\zeta_1 \in \{(x,y), (x,z), (y,z), (x,t), (y,t), (z,t)\}$ and $\zeta_2 \in \{(x,y,z,t)\}$. $\oplus$ refers to concatenate operation. $\text{interp}(\cdot)$ bilinearly samples the 3D Gaussian at the specified Hex-planes and diffusion feature-planes to obtain, respectively, the motion features and diffusion features. Diffusion features with tracking priors are extracted by projecting the Gaussian center $\mathcal{X}$ to the 2D diffusion feature plane using the camera's intrinsic matrix $K$ and extrinsic matrix $E$. To suppress erroneous supervision from self-occlusions, we perform ray-casting-based Glassner (1989) visibility checks on object-centric 2D images and discard occluded 3D-to-2D projections. We hypothesize that, under Stage One CoTracker supervision, the diffusion features learn motion priors that improve point representations of dynamic geometry and appearance for 4D reconstruction.

Our multi-head Gaussian deformation decoder $\phi$ consists of three lightweight MLP heads, each predicting the time-dependent offset for a Gaussian attribute such as position, rotation, and scale.

$$\Delta\mathcal{X} = \phi_{\mathcal{X}}(\mathcal{F}), \quad \Delta r = \phi_r(\mathcal{F}), \quad \Delta s = \phi_s(\mathcal{F}). \tag{7}$$

We design a 4D Spherical Harmonics (SH) appearance model by replacing each SH coefficient $k_l^m$ with a set of fourier transform coefficients $fr$ which are optimized as Gaussian attributes during the GS optimization stage. This models the appearance of a moving object as,

$$\mathcal{C}_{4D} = f_g(\psi, \gamma) = \sum_{l=0}^{l_{\max}} \sum_{m=-l}^{l} k_l^m Y_l^m(\psi, \gamma), \quad k_l^m = \sum_{i=0}^{w-1} fr_i \cos\left( \frac{i\pi}{N_t} t \right), \tag{8}$$

where $\mathcal{C}_{4D}$ is the predicted color for viewing direction $(\psi, \gamma)$ on the unit sphere; $Y_l^m(\psi, \gamma)$ denotes the real SH basis of degree $l$ and order $m$; $k_l^m$ are the corresponding SH coefficients; and $l_{\max}$ is the bandlimit controlling angular detail. The temporal parameterization $k_l^m$ models each coefficient over time, with $fr_i$ the optimized weights of a truncated cosine (Fourier) basis indexed by $i$, $N_t$ the total number of frames (setting the temporal frequency scale), and $w$ the number of retained cosine terms. Consequently, the canonical 4D Gaussian set is updated at time $t$ to yield the 4D Gaussians as follows:

$$\mathcal{G}_{4D} = \{\mathcal{X} + \Delta\mathcal{X},\ \mathcal{C}_{4D},\ \alpha,\ r + \Delta r,\ s + \Delta s\}, \tag{9}$$

The deformed 4D Gaussians are splatted onto the image plane to produce a set of 2D Gaussians. We then depth-sort the splats in camera space and apply standard $\alpha$-blending to accumulate per-pixel color, yielding the final rendered images $\hat{\mathcal{C}}$.

**Optimization Objectives:** We optimize the 4D-GS representation with three losses: motion reconstruction loss $\mathcal{L}_{\text{rec}}$, 4D-SDS loss $\mathcal{L}_{\text{4D−SDS}}$ Zhang et al. (2024); Shi et al. (2023); Jiang et al. (2024a), and ARAP loss $\mathcal{L}_{\text{ARAP}}$ Jiang et al. (2024a); Sorkine & Alexa (2007). Together, these terms first capture coarse motion from multi-view videos, then refine fine-grained motion via distillation, while stabilizing deformations with rigid movement constraints.

Using the multi-view videos produced in Stage One, we first reconstruct coarse motion by supervising 4D-GS renderings and masks over views and frames with a motion reconstruction loss $\mathcal{L}_{\text{rec}}$:

$$\mathcal{L}_{\text{rec}} = \frac{1}{nf} \sum_{i=1}^{n} \sum_{j=1}^{f} \left( \left\| \mathcal{M} \cdot \mathcal{C} - \hat{\mathcal{M}} \cdot \hat{\mathcal{C}} \right\|^2 \right), \tag{10}$$

where $\hat{\mathcal{C}}$ and $\mathcal{C}$ are the multi-view, multi-frame renderings and the corresponding ground truth, and $\hat{\mathcal{M}}$ and $\mathcal{M}$ are the object masks used to focus supervision on the foreground.

To model fine-grained motion, we distill the Stage One multi-view diffusion prior into the reconstructed 4D-GS using a $z_0$-reconstruction 4D-SDS loss $\mathcal{L}_{\text{4D-SDS}}$,

$$\mathcal{L}_{\text{4D−SDS}}\big(\mathcal{G}_{4D}, z = \mathcal{E}(\text{Render}(\mathcal{G}_{4D}))\big) = \mathbb{E}_{t,c,\epsilon}\left[ \left\| z - \hat{z}_0 \right\|_2^2 \right],\ \ \hat{z}_0 = \frac{z_t - \sigma_t \epsilon_\theta}{\alpha_t}, \tag{11}$$

where $z$ and $z_0$ are latent features of the rendered image and the estimation of latent feature from U-Net's current noise prediction $\epsilon_\theta$, respectively. $\sigma_t$ and $\alpha_t$ denote the noise scale and signal controlled by the noise scheduler, and $c$ represents the camera parameters. We also adopt the ARAP loss Jiang et al. (2024a) $\mathcal{L}_{\text{ARAP}}$ to facilitate the rigid movement learning. The final objective of Stage Two is $\mathcal{L}_2 = \lambda_4 \mathcal{L}_{\text{rec}} + \lambda_5 \mathcal{L}_{\text{4D−SDS}} + \lambda_6 \mathcal{L}_{\text{ARAP}}$ with $\lambda_4, \lambda_5, \lambda_6$ as weighting coefficients.

## 4 EXPERIMENTS

Following Animate3D Jiang et al. (2024a), we compare our method against baselines along two aspects: i) multi-view video generation, and ii) 4D generation.

**Training and Evaluation Datasets.** Multi-view video generation: We train on the MV-Video dataset from Animate3D (Sec. 4.1 in Jiang et al. (2024a)). Evaluation is conducted on two datasets: i) filtered Diffusion4D dataset (Sec. 3.2 in Liang et al. (2024)) and ii) Animate3D dataset (App. E.1 in Jiang et al. (2024a)). The filtered Diffusion4D dataset is derived from the original Diffusion4D dataset by removing all items that overlap with the Animate3D training set. After this filtering step, the resulting dataset contains 305 unique objects, which we use for our evaluations. 4D Generation: We evaluate on two datasets: (i) our reconstructed set of 28 4D assets from Sketchfab named *Sketchfab28* dataset, with processing details provided in Appendix D, and (ii) the 20 4D assets from the Animate3D Jiang et al. (2024a) dataset. Ablation study: We use a filtered Diffusion4D dataset for the multi-view video generation ablation study, and our Sketchfab28 dataset for the 4D generation ablation study.

**Evalaution Metrics:** Following VBench Huang et al. (2024), we evaluate multi-view video generation with five standard metrics that jointly capture consistency with image, appearance quality, motion quality, and aesthetics: 1) I2V, 2) Motion Smoothness (M. Sm); 3) Temporal Fidelity (T. Fli); 4) Dynamic Score (Dy. Sc). 5) Aesthetic Quality (Aest. Q); We evaluate 4D assets generation using both CLIP-based semantic alignment and a human user study. Following Zhang et al.

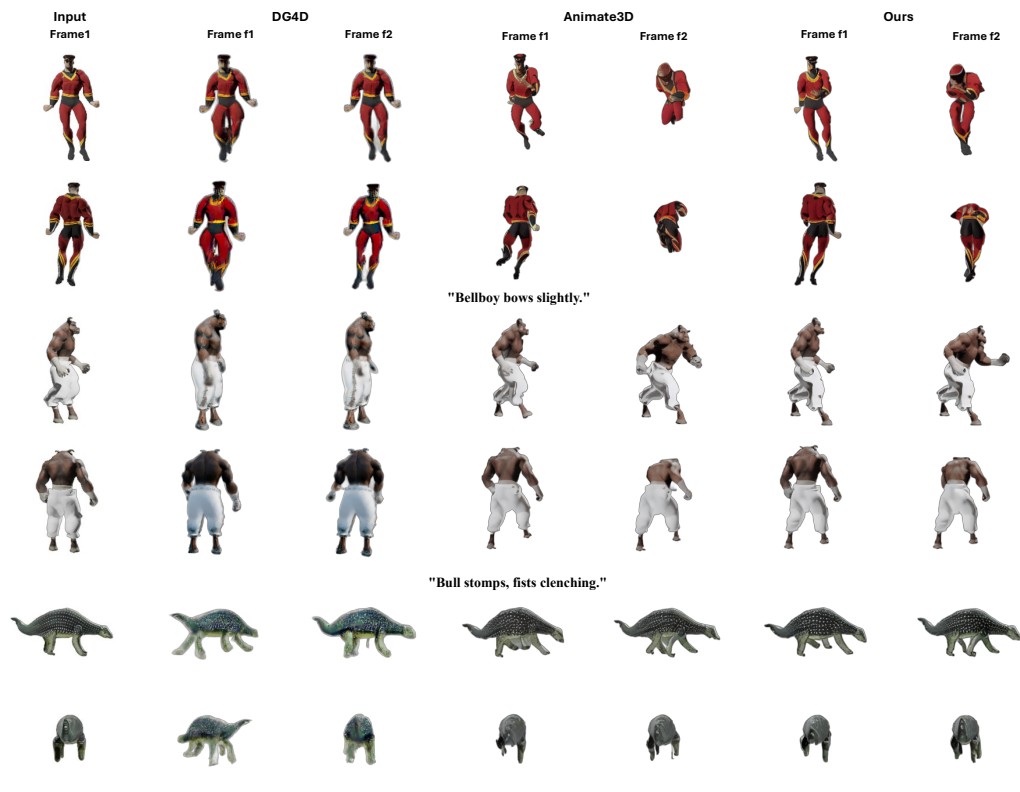

Figure 5: Video Generation qualitative comparison on Diffusion4D. Best viewed with zoom.

(2024); Wan et al. (2024), we report five CLIP-based metrics: 1) CLIP-O(img); 2) CLIP-O(text); 3) CLIP-F(img); 4) CLIP-F(text); (5) CLIP-C. In line with Animate3D Jiang et al. (2024a), we further conduct a user study with four criteria: 1) Alignment Text (Align. Text); 2) Alignment 3D (Align. 3D); 3) Motion (Mot.); (4) Appearance (Appr.). Precise definitions and computation details for all metrics are provided in the Appendix B.

**Implementation Details:** We provide the implementation details such as training/inference settings, architectures, and hyperparameters in Appendix A.

**Baseline Methods:** We benchmark *Track4Animate3D* against strong state-of-the-art baselines, including Animate3D Jiang et al. (2024a) and DG4D Ren et al. (2023). Our comprehensive evaluation encompasses quantitative metrics, qualitative visual comparisons, and a user study.

**Multi-view Video Generation Evaluation:** We provide video generation quantitative and qualitative comparisons for filtered Diffusion4D dataset in Tab. 1.left and Fig. 5, and for Animate3D dataset in Tab. 1.right and Fig. 6 (Appendix). On Tab. 1.left, our stage-one model achieves the best results on all five metrics, I2V 0.933 (+0.014 over Animate3D), M.Sm 0.992 (+0.001), T.Fli 0.991 (+0.002), Dy.Sc 1.356 (+0.008), and Aest.Q 0.470 (+0.005), indicating stronger image-to-video consistency, smoother motion, improved temporal stability, and higher aesthetic quality. On Tab. 1.right, our method ranks first on 4/5 metrics, I2V 0.945 (+0.018), M.Sm 0.993 (+0.002), T.Fli 0.992 (+0.002), Aest.Q 0.506 (+0.014), while trailing in Dy.Sc (0.778 vs. 0.787). As shown in the visual comparisons in Figs. 5 and 6 (Appendix), our method, leveraging foundation point-tracking motion priors, produces more appearance-consistent frames, more natural motion, and sharp object boundaries.

|  | I2V ↑ | M. Sm ↑ | T. Fli ↑ | Dy. Sc ↑ | Aest. Q ↑ |  | I2V ↑ | M. Sm ↑ | T. Fli ↑ | Dy. Sc ↑ | Aest. Q ↑ |
|---|---|---|---|---|---|---|---|---|---|---|---|
| DG4D | 0.834 | 0.983 | 0.982 | 1.019 | 0.445 | DG4D | 0.850 | 0.985 | 0.984 | 0.702 | 0.481 |
| Animate3D | 0.919 | 0.991 | 0.989 | 1.348 | 0.465 | Animate3D | 0.927 | 0.991 | 0.990 | **0.787** | 0.492 |
| Ours | **0.933** | **0.992** | **0.991** | **1.356** | **0.470** | Ours | **0.945** | **0.993** | **0.992** | 0.778 | **0.506** |

Table 1: Video Generation quantitative comparison on: Diffusion4D (left); Animate3D (right).

**4D Generation Evaluation:** We evaluate 4D generation using CLIP-based semantic alignment metrics and a controlled human user study. We provide 4D generation quantitative and qualitative comparisons for Sketchfab28 dataset in Tab. 2.top and Fig. 7 (Appendix), and for Animate3D dataset

in Tab. 2.bottom and Fig. 8 (Appendix). On Tab. 2.top, our stage-two model improves over the strongest baseline (Animate3D) by +0.0072 CLIP-O(img), +0.0011 CLIP-O(text), +0.0049 CLIP-F(img), +0.0030 CLIP-F(text), and +0.0018 CLIP-C. On Tab. 2.bottom, it yields +0.0044 CLIP-O(img), +0.0012 CLIP-O(text), +0.0000 (tie) CLIP-F(img), +0.0002 CLIP-F(text), and +0.0036 CLIP-C. These consistent gains in both object- and frame-level alignment (image/text) translate into a higher composite score, confirming better view–time fidelity and semantic faithfulness. As shown in Figs. 7 (Appendix) and 8 (Appendix), our method, leveraging diffusion features that encode tracking priors and 4D SH modeling, yields more accurate shapes, more realistic colors and lighting, and fewer artifacts.

| | CLIP-O(img) ↑ | CLIP-O(text) ↑ | CLIP-F(img) ↑ | CLIP-F(text) ↑ | CLIP-C ↑ |
|---|---|---|---|---|---|
| DG4D | 0.8619 | 0.2578 | 0.8708 | 0.2592 | 0.9700 |
| Animate3D | 0.8812 | 0.2653 | 0.8906 | 0.2634 | 0.9801 |
| Ours | **0.8884** | **0.2664** | **0.8955** | **0.2664** | **0.9819** |
| DG4D | 0.8632 | 0.2897 | 0.8836 | 0.2940 | 0.9761 |
| Animate3D | 0.9384 | 0.2968 | 0.9447 | 0.3003 | 0.9808 |
| Ours | **0.9428** | **0.2980** | **0.9447** | **0.3005** | **0.9844** |

Table 2: 4D Generation quantitative comparison on: Sketchfab28 (top); Animate3D (bottom).

Align with Animate3D Jiang et al. (2024a), we conduct a user study for 4D generation with 20 participants on our Sketchfab28 and Animate3D datasets, reporting mean scores in Tab. 3. Participants rate each generated dy-

| | Align. Text ↑ | Align. 3D ↑ | Mot. ↑ | Appr. ↑ |
|---|---|---|---|---|
| DG4D | 3.38 | 3.73 | 3.37 | 3.27 |
| Animate3d | 3.50 | 3.78 | 3.63 | 3.41 |
| Ours | **3.57** | **3.87** | **3.73** | **3.44** |

Table 3: 4D Generation User study.

namic object on a 1–5 scale (1-lowest, 5-highest) for alignment with the text prompt-Alignment Text (Align. Text), alignment with the static asset-Alignment 3D (Align. 3D), motion quality-Motion (Mot.), and appearance quality-Appearance (Appr.). The final score for each metric is the average of all its ratings. The results consistently favor our approach, indicating superior perceptual alignment and visual quality.

**Ablation Study:** Following Animate3D Jiang et al. (2024a), we assess the contribution of each module in *Track4Animate3D* via controlled ablations for multi-view video generation and 4D generation. For multi-video generation, we compare three variants: (1) w/o Corrs. Loss (correspondence loss), (2) w/o pos. Loss (position loss), and (3) full model. For 4D generation, we also compare three variants: (1) w/o Di. Feat (diffusion features), (2) w/o 4D SH, and (3) the full model. Quantitative and qualitative results are reported in Tab. 4.left and Fig. 9 (Appendix) for video generation, and in Tab. 4.right and Fig. 10 (Appendix) for 4D generation. Removing correspondence/position supervision weakens multi-view temporal coherence in video generation, while dropping diffusion features or the 4D SH degrades appearance fidelity and increases artifacts in 4D generation, confirming each component's complementary benefit.

| | I2V ↑ | M. Sm ↑ | T. Fli ↑ | Dy. Sc ↑ | Aest. Q ↑ | | I2V ↑ | M. Sm ↑ | T. Fli ↑ | Dy. Sc ↑ | Aest. Q ↑ |
|---|---|---|---|---|---|---|---|---|---|---|---|
| w/o Corrs. Loss | 0.844 | 0.990 | 0.990 | **1.505** | 0.347 | w/o Di. Feat | 0.932 | 0.994 | 0.990 | **1.292** | 0.532 |
| w/o Pos. Loss | 0.921 | 0.991 | 0.989 | 1.435 | 0.462 | w/o 4D SH | 0.937 | 0.995 | 0.993 | 1.290 | 0.536 |
| Ours full | **0.933** | **0.992** | **0.991** | 1.356 | **0.470** | Ours full | **0.940** | **0.996** | **0.994** | 1.286 | **0.538** |

Table 4: Ablation studies: Video Generation (left); 4D Generation (right).

**Comprehensive Visualizations.** We provide comprehensive qualitative comparisons for both video generation and 4D generation in Appendix C. We also provide the ablation study visualization for both video generation and 4D generation in Appendix C. We attach interactive HTML video demos in Appendix F for a more intuitive visualization of appearance and motion.

## 5 CONCLUSION

We present *Track4Animate3D*, a two-stage framework that unifies multi-view video diffusion, point tracker, and 4D-GS to animate arbitrary static 3D models. The core idea is to leverage foundation trackers as motion experts and inject their knowledge into both video generation and 4D-GS. By injecting dense, feature-level point correspondences into the diffusion generator, our approach imposes explicit correspondence-aware supervision that suppresses appearance drift and strengthens temporal coherence for multi-view video generation. On the reconstruction side, we introduce a 4D-GS pipeline with a hybrid representation that concatenates co-located diffusion features carrying tracking priors with standard Hex-plane features, and design a 4D SH modeling, yielding higher-fidelity dynamics and illumination. Together, these motion cues enhance temporal alignment, handle complex motion, and maintain consistent feature representations over time.

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

# Appendix

Figure 6: Video Generation qualitative comparison on Animate3D. Best viewed by zooming in.

## A   IMPLEMENTATION DETAILS

For multi-view video diffusion training, we uniformly sample 16 frames per animation. We optimize with AdamW (learning rate 1e-6, weight decay 0.01) using a constant-with-warmup LR scheduler for 20 epochs with a batch size of 1024 (16 × 4 views × 16 frames) on 16 NVIDIA H100 GPUs. At inference, we use 25 sampling steps to obtain stable 3D object animations. For 4D reconstruction, the Hex-plane resolution and feature dimension are set to [100, 100, 8] and 16, respectively. We perform motion reconstruction progressively for the first 750 iterations with a batch size of 64 (4 views × 16 frames), and then enable 4D-SDS and ARAP for an additional 250 iterations. The learning rates are 0.01 for Hex-planes and 1e-4 for the offset prediction layers. The loss weights $\lambda_1, \lambda_2, \lambda_3, \lambda_4, \lambda_5, \lambda_6$ are set to 1.0, 0.1, 10, 100.0, 0.01 and 10.0 respectively. Training the multi-view diffusion model takes around 12-14 hours on 16 × 80GB H100 GPUs. Per-object 4D optimization takes 40 minutes on a single A100 GPU (around 20 minutes for motion reconstruction and around 20 minutes for 4D-SDS and ARAP).

## B   EVALAUTION METRICS DETAILS

**Multi-view Video Generation Evaluation Metrics.** Following VBench Huang et al. (2024), we evaluate multi-view video generation with five complementary metrics that jointly capture appearance/semantic fidelity, motion quality, and aesthetics: 1) I2V: image-to-video consistency computed in the DINO feature space, quantifying appearance alignment between the conditioning image and the generated video; 2) Motion Smoothness (M.,Sm): temporal smoothness estimated by comparing

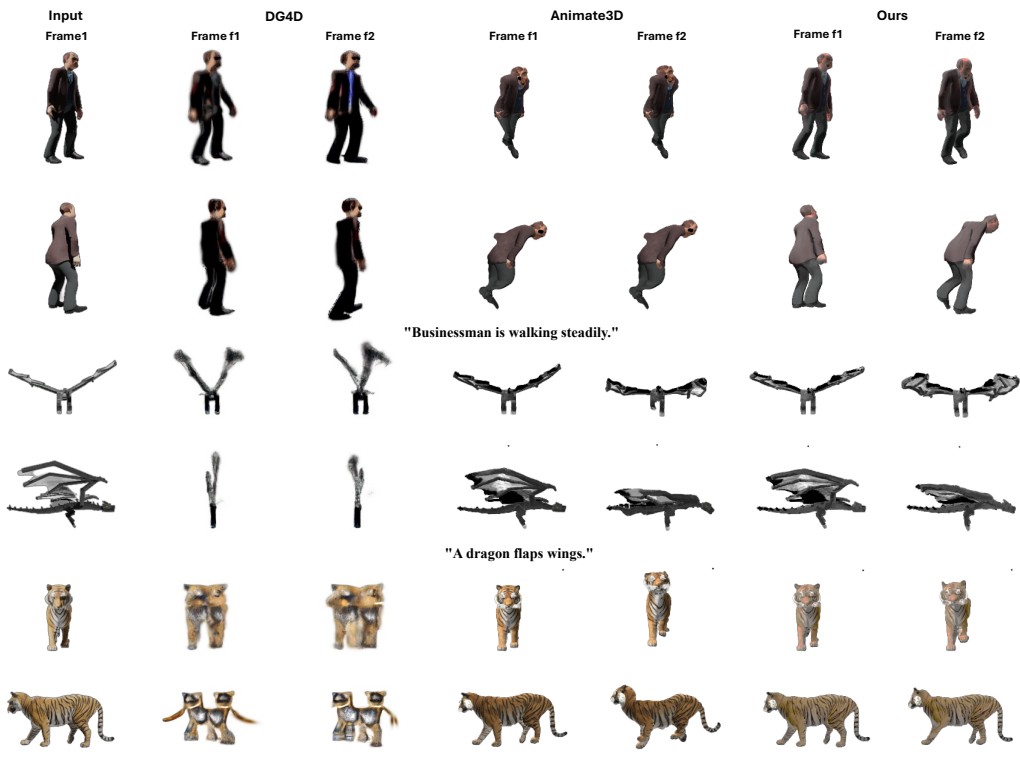

Figure 7: 4D Generation qualitative comparison on Sketchfab28. Best viewed by zooming in.

the generated sequence with frames synthesized via interpolation, penalizing jitter and discontinuities across time; 3) Temporal Fidelity (T.,Fli): tability of scene content under viewpoint changes, measuring how well identities, geometry, and textures persist over the sequence; 4) Dynamic Score (Dy.,Sc): a continuous estimate of motion intensity derived from the same flow-based predictor, reflecting the richness and realism of dynamics without over- or under-motion; 5) Aesthetic Quality (Aest.,Q):perceptual appeal scored by the LAION Aesthetic Predictor, capturing overall visual quality of rendered frames.

**4D Generation Evaluation Metrics.** Our evaluation of 4D asset generation includes CLIP-based semantic alignment and a human user study.

Following prior work Zhang et al. (2024); Wan et al. (2024), we evaluate 4D-GS reconstructions with five CLIP-based similarity metrics computed in the CLIP embedding space, aggregating over views/frames as specified below: 1) CLIP-O(img): CLIP similarity between the rendered object image and the reference static object image; 2) CLIP-O(text): CLIP similarity between the rendered object image and the text prompt describing that object; 3) CLIP-F(img): frame-wise CLIP similarity between rendered video frames and the reference static object image; 4) CLIP-F(text): frame-wise CLIP similarity between rendered video frames and the text prompt; 5) CLIP-C: composite score aggregating object-/frame-level, image-/text-conditioned similarities.

Following prior work in Animate3D Jiang et al. (2024a), we further conduct a user study evaluated along four different criteria: 1) Alignment Text (Align. Text). Measures how closely the generated 4D asset (appearance + motion) semantically matches the input prompt-covering identity, attributes (color/material/parts), and action semantics. Raters assign 1-5 Likert scores (higher is better). 2) Alignment 3D (Align. 3D). Assesses geometric and canonical look consistency between the generated 4D asset and the provided static reference, emphasizing shape fidelity, fine textures, and identity preservation across views/time. Scored 1-5. 3) Motion (Mot.). Evaluates perceived motion quality, including physical plausibility, temporal smoothness, and continuity of articulations, while penalizing jitter, ghosting, or flicker. Scored 1-5. 4) Appearance (Appr.). Rates per-frame visual fidelity and aesthetics-sharpness, texture richness, material/lighting realism-and penalizes artifacts such as blur, seams, or ringing. Scored 1-5.

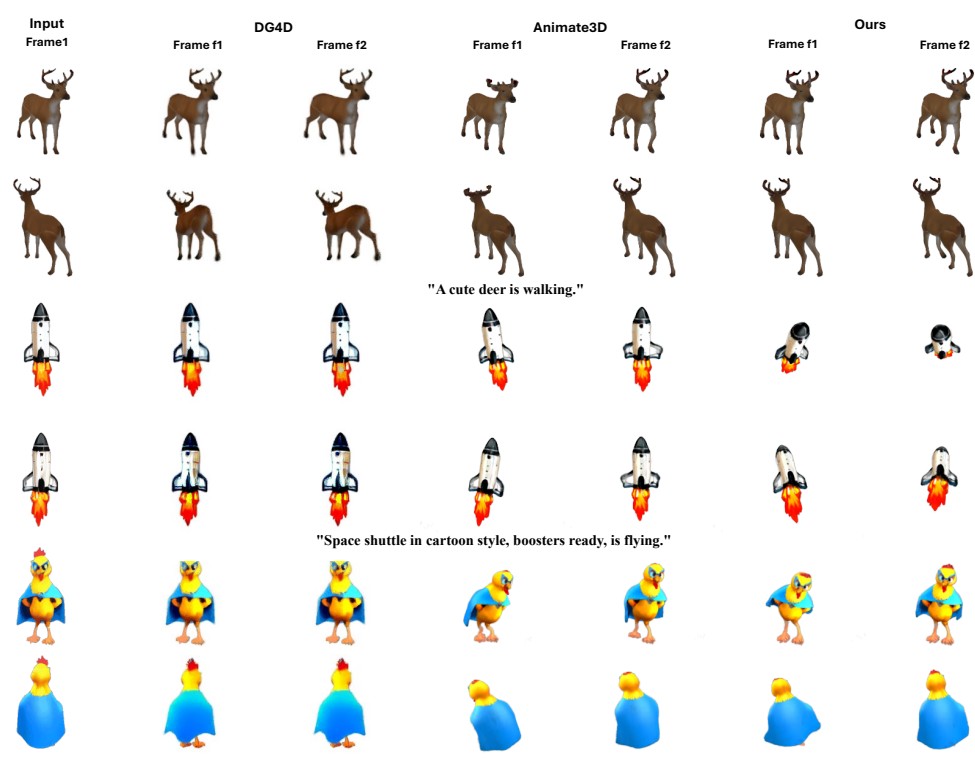

Figure 8: 4D Generation qualitative comparison on Animate3D. Best viewed by zooming in.

## C    MORE VISUALIZATIONS

We provide all the qualitative visualization results for both video generation and 4D generation in Figs. 6, 7 and 8, covering diverse objects, viewpoints, and motion regimes. Please note that for some DG4D examples, the rendered view is not perfectly aligned with the input view; for instance, the Spider-Man result in the second row of Fig. 6. This mismatch arises from object position movement introduced by the DG4D codebase during rendering. We also provide ablation visualizations for both video generation and 4D generation in Figs. 9 and 10.

## D    MORE DETAILS ON NEW EVALUATION DATASET

This section details the construction of our new evaluation dataset, *Sketchfab28* dataset. We processed 28 object meshes by baking realistic lighting directly into their vertex colors. To achieve consistent and high-quality illumination, we used a standard four-point lighting setup in Blender, consisting of a key light, fill light, back light, and background light. The key light provided the primary illumination to define the object's shape, while the fill light softened shadows, the back light enhanced depth by separating the object from the background, and the background light ensured even lighting of the scene. During baking, both direct and indirect light contributions were included, with all major shading components enabled, such as diffuse, glossy, transmission, and emission. This allowed the baked vertex colors to capture realistic shading and reflectance effects without relying on external texture maps. The baked meshes were then converted into 3D Gaussian splats, which serve as the input representation for subsequent stages of our pipeline. We will release the *Sketchfab28* dataset for futher research.

## E    USER STUDY TEMPLATE

As shown in Fig. 11, we present the interface used in our user study. The survey evaluates the 4D objects from Sketchfab28 and Animate3D datasets. For each generated 4D object, participants

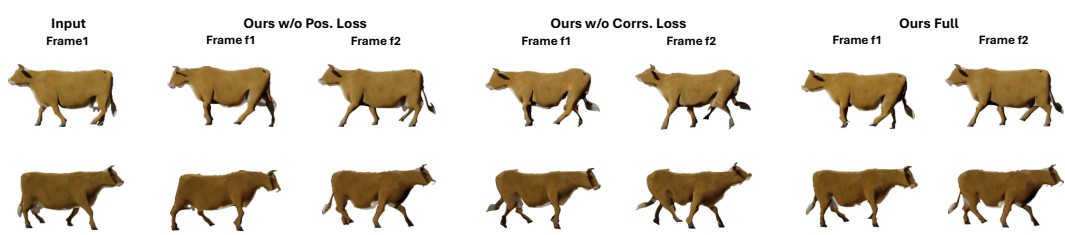

Figure 9: Video Generation Ablation. Best viewed by zooming in.

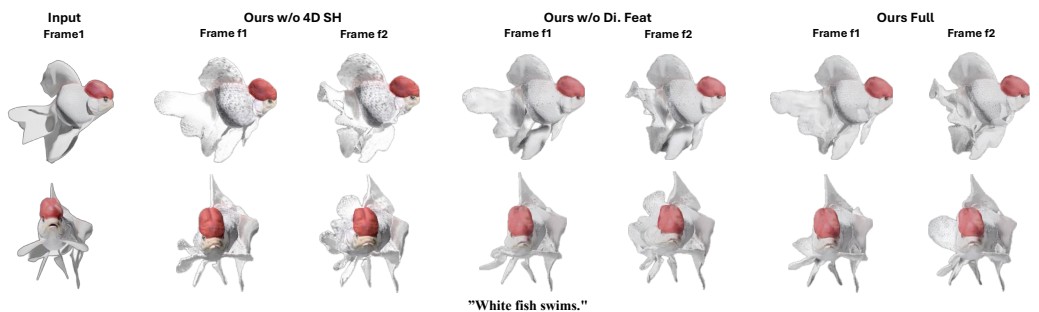

Figure 10: 4D Generation Ablation. Best viewed by zooming in.

provide 1–5 Likert ratings on three criteria: alignment with the given static object and text prompt, appearance fidelity, and motion quality.

## F  VIDEO DEMOS

We provide interactive HTML demos for i) generated 4D assets and ii) multi-view point tracking, enabling synchronized multi-view comparisons. For the generated 4D assets, the early test cases are relatively simple, where our method slightly outperforms Animate3D and clearly surpasses DG4D. The later, substantially more challenging cases show our approach significantly outperforming both baselines.

**Usage:**

1. Download and unzip
   `10559_Track4Animate3D_Animatin_SupplementaryMaterial.zip`.
2. Open `Generated_4D_Assets/Generated_4D_Assets.html` in a web browser to view the generated 4D asset demos.
3. Open `Point_Tracking/Point_Tracking.html` in a web browser to view the multi-view point-tracking demos.
4. *Do not* modify the extracted folder structure or relative file paths; the HTML pages rely on them to load assets correctly.

## G  REPRODUCIBILITY STATEMENT

To facilitate faithful replication, we will release, upon acceptance, the full codebase, all trained checkpoints, the evaluation toolkit, and the dataset.

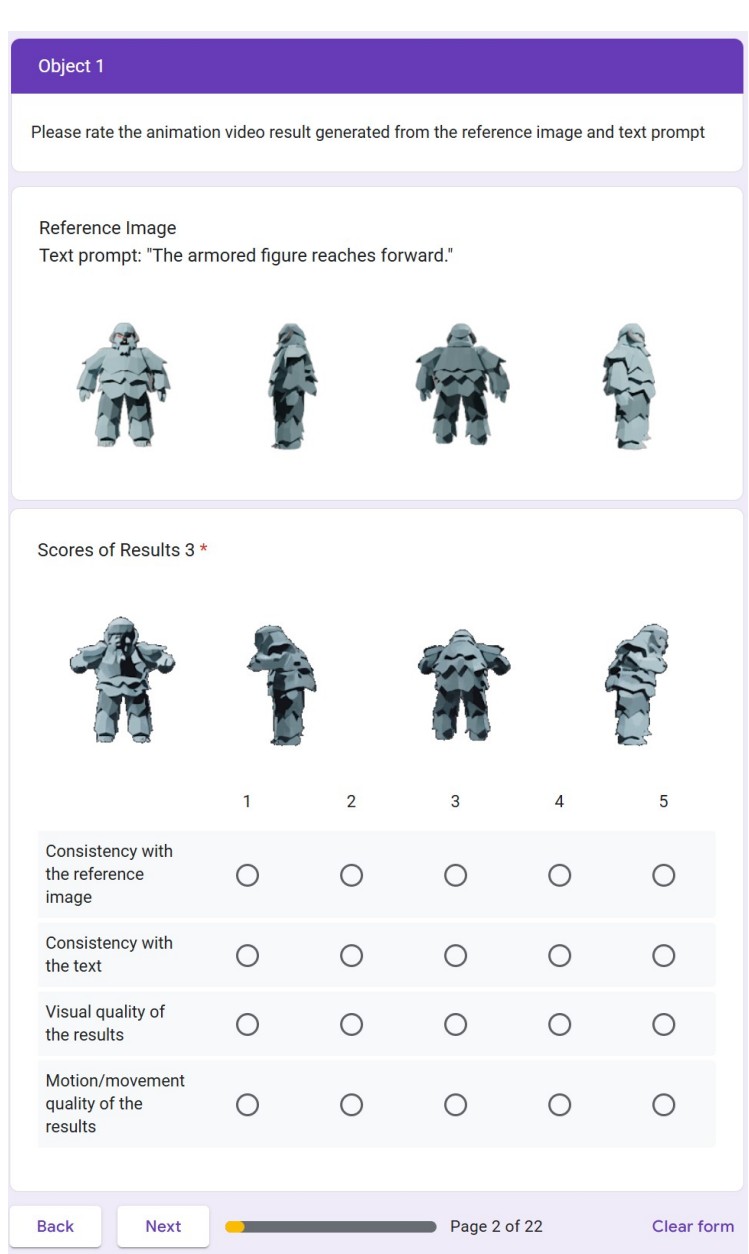

Figure 11: The UI of our user study.

