# OpenReview forum: "Track4Animate3D: Animating Any 3D Model via Multi-View Diffusion with Point-Tracking Motion Priors"
_ICLR.cc/2026/Conference — ICLR 2026 Conference Withdrawn Submission_

### Official Review · Reviewer_J3y3 · 2025-10-31

**Soundness:** 3
**Presentation:** 3
**Contribution:** 3
**Rating:** 6
**Confidence:** 4

**Summary:**

This paper introduces Track4Animate3D, a novel two-stage framework designed to solve the core challenges in 4D object generation—such as maintaining appearance and motion consistency over time and avoiding visual artifacts.

**Strengths:**

1. The paper proposes a 4D generation framework leveraging motion priors to mitigate appearance drift through explicit temporal supervision.
2. By introducing motion priors into multi-view video generation and 4DGS reconstruction, the proposed approach effectively enhances temporal consistency across frames.
3. The authors construct a new high-quality 4D dataset, which provides valuable resources for future research in dynamic 3D and 4D generation.

**Weaknesses:**

1. The entire framework heavily relies on the motion priors provided by CoTracker3. In practice, point tracking often fails under occlusion or textureless regions. How does the method handle those situation to maintain robustness?
2. The paper would benefit from a runtime efficiency comparison (e.g., training or inference time) against baselines such as Animate3D, to demonstrate that the proposed approach is computationally feasible.
3. The authors should briefly justify why the chosen intermediate-layer features are more suitable for point tracking compared to shallower or deeper layers (e.g., mid-block or final upsampling block).

**Questions:**

see weaknesses

---

### Official Review · Reviewer_W42A · 2025-11-01

**Soundness:** 2
**Presentation:** 2
**Contribution:** 2
**Rating:** 4
**Confidence:** 4

**Summary:**

The paper tackles 3D-to-4D asset animation by combining a multi-view video diffusion backbone with dense point-tracking supervision and a hybrid 4D Gaussian splatting reconstructor. Stage One injects CoTracker3-derived correspondences into the diffusion U-Net, identifying a stable feature layer for tracking-aware losses that aim to curb appearance drift. Stage Two concatenates these diffusion features with Hex-plane encodings and extends appearance modeling with 4D spherical harmonics while optimizing with reconstruction, 4D-SDS, and ARAP losses.

**Strengths:**

- The paper makes a compelling observation that existing multi-view video diffusion approaches suffer from appearance drift due to supervision limited to pixel/latent space. The solution of leveraging CoTracker3 to inject dense feature-level correspondences into the diffusion U-Net is intuitive and well-motivated.
- The curation of Sketchfab28 with baked lighting addresses a real need for object-centric 4D evaluation data.

**Weaknesses:**

- In the supplementary videos comparing with Animate3D and DG4D, I don’t see a clear improvement over Animate3D in temporal consistency or visual quality. Why is that?
- The paper does not include a comparison with recent SOTA methods such as AnimateAnyMesh [1].
- Incorporating point tracking into multi-view diffusion is interesting, but the overall pipeline feels incremental. It mainly combines existing techniques (multi-view diffusion, point tracking, and 4D Gaussian splatting) without introducing a fundamentally new idea.

[1] AnimateAnyMesh: A Feed-Forward 4D Foundation Model for Text-Driven Universal Mesh Animation. In Proceedings of the IEEE/CVF International Conference on Computer Vision (ICCV)

**Questions:**

- What are the end-to-end training and inference times, and the GPU memory usage for each stage, compared to Animate3D and AnimateAnyMesh?

---

### Official Review · Reviewer_RREp · 2025-11-01

**Soundness:** 2
**Presentation:** 3
**Contribution:** 2
**Rating:** 4
**Confidence:** 5

**Summary:**

This paper proposes a new two-stage 4D generation framework, Track4Animate3D, which aims to improve the temporal consistency of multi-view video generation by introducing motion priors from a foundation point tracker, and to enhance dynamic reconstruction quality based on 4D Gaussian Splatting. In the first stage, it injects dense, feature-level supervision from CoTracker3 into a multi-view video diffusion model, using designed Losses to explicitly enforce temporal consistency and thereby mitigate appearance drift. In the second stage, diffusion features that carry tracking priors are concatenated with Hex-plane features to capture dynamic geometry and illumination changes during 4D-GS reconstruction. The authors conduct extensive experiments on multiple datasets and introduce a new evaluation set, Sketchfab28.

**Strengths:**

1.	The motivation of utilizing pre-trained tracking models to facilitate the spatially and temporally consistent feature learning in multi-view video generation pipeline is novel.

2.	The paper is well-organized and easy to follow

3.	The authors conduct extensive experiments to validate their design for video generation and 4D reconstruction.

**Weaknesses:**

1.	Trajectory ground truth: For trajectory ground truth in training dataset, the authors use CoTracker to predict the trajectory. However, under under rapid deformation, heavy occlusion, or homogeneous textures, CoTracker might output wrong results. A more proper way is to extract ground truth trajectory from original mesh files. Besides, the authors didn’t mention the uncertainty modeling for tracking noise and occlusions, which might force wrong correspondence learning when finetuning video diffusion.

2.	The contribution of tracking supervision to cross-view consistency is limited. In the paper, CoTracker is run independently for each view, which can ensure temporal continuity within a single view but not cross-view geometric correspondence. Although temporal supervision is introduced at the feature level, it mainly strengthens within-view stability rather than establishing a unified representation across views. The authors are suggested to evaluate the contribution of tracking supervision within-views and cross-views separately.

3.	Evaluation dataset: the Diffusion4D test dataset might have overlap with Animate3D training dataset, cause they all downloaded from sketchfab. The authors are suggested to check this carefully.

4.	 Comparison method: DG4D is not a multi-view video generation work, so it is inappropriate to compare multi-view video generation with it. The authors are suggested to compare with Diffusion4D/4Diffusion/SV4D/SV4D++. For 4D generation, please add L4GM aside from those in multi-view video generation.

**Questions:**

Static 3D-4D has some new methods, such as AnimateAnyMesh and DriveAnyMesh. The authors might could consider discuss them in related work.
I am willing to give a higher recommendation if the authors solve my concerns.

---

### Note · Authors · 2025-11-13

I have read and agree with the venue's withdrawal policy on behalf of myself and my co-authors.